

# Differentially expressed genes related to plant height and yield in two alfalfa cultivars based on RNA-seq

Jiangjiao Qi, Xue Yu, Xuzhe Wang, Fanfan Zhang and Chunhui Ma

College of Animal Science & Technology, Shihezi University, Shihezi, Xinjiang, China

## ABSTRACT

**Background**. Alfalfa (*Medicago sativa* L.) is a kind of forage with high relative feeding value in farming and livestock breeding, and is of great significance to the development of animal husbandry. The growth of the aboveground part of alfalfa is an important factor that limits crop yield. Clarifying the molecular mechanisms that maintain vigorous growth in alfalfa may contribute to the development of molecular breeding for this crop.

**Methods**. Here, we evaluated the growth phenotypes of five cultivars of alfalfa (WL 712, WL 525HQ, Victoria, Knight 2, and Aohan). Then RNA-seq was performed on the stems of WL 712, chosen as a fast growing cultivar, and Aohan, chosen as a slow growing cultivar. GO enrichment analysis was conducted on all differentially expressed genes (DEGs).

**Result**. Among the differentially expressed genes that were up-regulated in the fast growing cultivar, GO analysis revealed enrichment in the following seven categories: formation of water-conducting tissue in vascular plants, biosynthesis and degradation of lignin, formation of the primary or secondary cell wall, cell enlargement and plant growth, cell division and shoot initiation, stem growth and induced germination, and cell elongation. KEGG analysis showed that differentially expressed genes were annotated as being involved in plant hormone signal transduction, photosynthesis, and phenylpropanoid biosynthesis. KEGG analysis also showed that up-regulated in the fast growing cultivar were members of the *WRKY* family of transcription factors related to plant growth and development, members of the *NAC* and *MYB* gene families related to the synthesis of cellulose and hemicellulose, and the development of secondary cell wall fibres, and finally, *MYB* family members that are involved in plant growth regulation. Our research results not only enrich the transcriptome database of alfalfa, but also provide valuable information for explaining the molecular mechanism of fast growth, and can provide reference for the production of alfalfa.

## INTRODUCTION

The stem is an important vegetative organ between the root and leaf of a plant and transports nutrients and water (*Aliche et al., 2020*). The stems of alfalfa also play a role in photosynthesis, nutrient storage, and regeneration (*Perez-Garcia & Moreno-Risueno, 2018*). In the process of stem growth and development, stem tips grow continuously, whereas

Corresponding author
Chunhui Ma, chunhuima@126.com

branches, leaves, and lateral branches are produced successively, which together constitute a huge branch system (*Yu et al., 2015*; *Jaykumar & Mahendra, 2016*). The degree of stem development is closely related to the life cycle of plants (*Etzold et al., 2021*), especially the aboveground biomass of the plant (*Kleyer et al., 2019*). Alfalfa, with stems and branches as the main components of biomass yield, is a typical representative crop.

Alfalfa is a feed crop with a high economic value (*Kumar et al., 2018*). In addition to its stress resistance properties, it has been the focus of research because of its perennial nature and high nutritional value (*Wang et al., 2017*; *Diatta, Doohong & Jagadish, 2021*). The stems and leaves of alfalfa have a high nutrient content and are the main parts areas of animal forage (*Sulc et al., 2021*). Owing to the cross-pollination of alfalfa, most cultivars have a complex genetic background. Restricted by its genetic characteristics, growth performance and nutritional quality are uneven (*Bambang et al., 2021*). Alfalfa stalks are composed of nodes and internodes, which affect plant height and yield. The height and stem diameter of alfalfa are important factors that restrict its biomass (*Monirifar, 2011*). Therefore, increasing the number of alfalfa vegetative branches, vegetative growth time, and delaying the flowering time of plants are crucial for improving the nutritional quality and yield of forage grass (*Aung et al., 2015*).

Previous studies have reported significant differences in alfalfa plant height and biomass yield among cultivars (*Ziliotto et al., 2010*). The series of WL alfalfa cultivars had the best growth performance when compared among cultivars (*Tetteh & Bonsu, 1997*). Plant spacing and light significantly effect alfalfa forage yield and weed inhibition (*Celebi et al., 2010*). Compound fertilizers can increase the nutrient content of soil and improve the yield of alfalfa (*McDonald, Baral & Min, 2021*; *Na et al., 2021*). Additionally, the growth and development periods of alfalfa are equally important for its yield (*Martin et al., 2010*). During the growth of alfalfa, the budding stage, which has excellent nutritional quality and biomass yield, has always been a period of concern for breeders (*Fan et al., 2018*). Currently, research on the growth performance of alfalfa mainly focuses on the physiological level. Few reports have revealed the molecular mechanism of alfalfa stem elongation and diameter enlargement. Owing to the lack of a complete reference genome sequence, previous studies on the stress- response genes of alfalfa have used nonparametric transcriptome analysis (*Yuan et al., 2020*; *Wang et al., 2021*; *Gao et al., 2016*; *Arshad, Gruber & Hannoufa, 2018*). Reference-free transcriptome refers to the sequencing of eukaryotic transcriptomes in the absence of a reference genome. After obtaining the original data for eukaryotic nonparametric transcriptome sequencing, the quality control splicing is first performed to generate unigenes, which are then used as the reference sequence for subsequent analysis. However, with the availability of whole- genome sequencing and annotation of alfalfa (Zhongmu 1), studying the alfalfa genome has become easier (*Zhang et al., 2021*).

Transcriptome sequencing is the study of all mRNAs present in a given sample, which is the basis for the study of gene function and is important for understanding the development of organisms. With the advantages of high-throughput, high accuracy, and high sensitivity, RNA- seq can be used to study changes in the expression level of transcripts to understand or reveal the intrinsic relationship between gene expression and biological phenotypes. At present, RNA- seq technology has become a common method to study the growth

and development of many plants (*Chen et al., 2020*; *Kim et al., 2021*; *Zheng et al., 2021*). Next-generation high-throughput sequencing technology can be used to comprehensively obtain the transcript information of alfalfa and screen out genes related specifically to stem elongation and diameter enlargement.

The growth rate of alfalfa is an important factor that affects plant height and yield (*Yan et al., 2021*). Exploring the molecular mechanisms in alfalfa that regulate growth rate may be helpful to improve yield. Here, we identified differentially expressed genes (DEGs) in the stem of alfalfa "WL 712" (USA, Fall Dormancy = 10.2) and "Aohan" (China, Fall Dormancy = 2.0) using RNA-seq. We further identified key genes influencing vigorous-growing alfalfa by bioinformatics analysis and predicted their functions. These results may be helpful in clarifying the molecular mechanism that regulate growth rate in alfalfa, establishing a regulatory network of the growth and development of dominant cultivars, and laying a theoretical foundation for molecular breeding and the introduction of productive cultivars.

## MATERIALS & METHODS

### Characterisation of phenotypic traits

Five cultivars of alfalfa, *Medicago sativa* L. (WL 712, Victoria, WL 525HQ, Knight 2, and Aohan) were planted at the experimental station of Shihezi University, Xinjiang, China (N44°20′, E88°30′, altitude 420 m) (Table S1A). Its characteristic is temperate continental arid climate, with an average annual temperature of 8.1 °C. Before planting, we adopted the "S" shaped sampling method, and nine soil samples were obtained. The nutrient status of the soil (20 cm) was as follows: available nitrogen 92.6 mg/kg, organic matter 12.4 g/kg, available potassium 168.5 mg/kg, available phosphorus 33.2 mg/kg, and pH 7.26 (Table S1B).

In June 2019 and 2020, alfalfa was planted in a 40 m$^2$ plot using a completely randomised design. To ensure consistency among the cultivars, thirty-six stems were collected from a well- growing single plant of each cultivar. The single-row planting method was used with sampling plant spacing of 40 cm and row spacing of 60 cm, with three biological replicates per cultivar.

At the budding stage, agronomic traits of five randomly selected plants were determined from each of the three biological replicates. The absolute distance from the root to the top of the main stem was measured as plant height by using a ruler. The number of branches and nodes was counted. The stem diameter and internode length were measured by using calipers. The leaf area was measured by using a leaf area meter. Five plants in each row were randomly selected and weighed, and the average value was calculated as the total fresh weight per plant. By comparing and analyzing the growth indexes of different varieties, it was finally determined that WL 712 represented a vigorous and fast growing variety and Aohan represented a short and slow growing variety (Fig. 1).

### Cultivation of experimental materials and sample collection

Stems of WL 712 and Aohan were collected and cut into eight cm pieces, leaving an axillary bud. The stems were cultivated on cutting beds in the greenhouse (light/dark: 16 h/8 h,

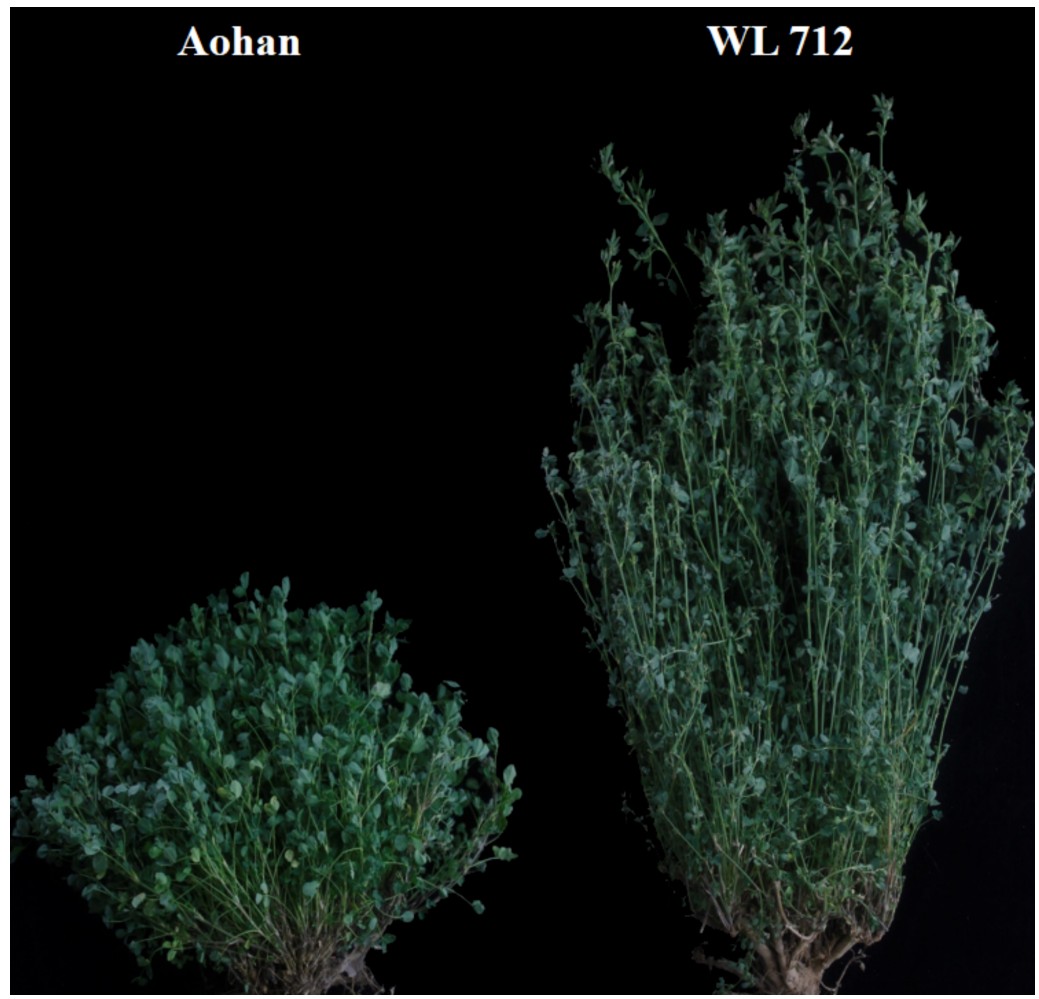

**Figure 1  Slow-growing Aohan and vigorous-growing WL 712 plants at bud stage.** Soil grown plants, approximately 60 days after planting.

Temp: 25 °C/20 °C, humidity 70%) of the Beiyuan campus of Shihezi University for 20 days, and surviving plants were transplanted into plastic pots (diameter 32 cm, height 35 cm). Nutrient soil: vermiculite = 1: 1 (cultivation and management methods were consistent). More than 30 individual plants of both WL 712 and Aohan survived in the greenhouse. Five plants each of WL 712 and Aohan alfalfa were randomly selected and the plant height, internode length, stem diameter, leaf area and yield were determined.

At the budding stage (about 42 days after transplanting), the plant height of WL 712 and Aohan reached 50.2 cm and 28.7 cm (Table 1), respectively. We collected the main stem and removed its top and base. The middle part of the main stem (approximately 1.5 cm in length) of each cultivar was collected, quickly frozen in liquid nitrogen. Three biological replicates were used for per cultivar. WJ1, WJ2 and WJ3 represent samples from the WL 712 cultivar. AJ1, AJ2 and AJ3 represent samples from the Aohan cultivar. Finally, six samples were used for RNA-seq.

**Table 1  The growth index of the two varieties in greenhouse-grown plants.**

|  | Plant height (cm) | Lenght of internodde (cm) | Stem diameter (mm) | Leaf areas (mm²) | Plant weight (g/plant) |
|---|---|---|---|---|---|
| WL 712 | $50.2 \pm 1^a$ | $5.14 \pm 0.09^a$ | $2.52 \pm 0.022^a$ | $159 \pm 0.6^a$ | $231 \pm 2.4^a$ |
| Aohan | $28.7 \pm 1^c$ | $2.94 \pm 0.07^c$ | $1.19 \pm 0.027^c$ | $127 \pm 2.8^c$ | $141 \pm 0.4^c$ |

Notes.
Different letters indicate significant difference at $P < 0.05$ among the two varieties as determined by Student's t test.

## Library construction and RNA-seq

Total RNA was isolated from stems using the RNeasy Plant Mini Kit (Qiagen, Hilden, Germany). A total of 3 µg RNA per sample was used to build the library. Sequencing libraries were generated using a NEBNext Ultra RNA Library Prep Kit (NEB, Ipswich, MA, USA). Messenger RNA was 1purified from each sample using magnetic beads and fragmented with divalent cations at elevated temperature. First-strand cDNA was obtained using segmented mRNA as template and random oligonucleotide as primer. Then, the second strand of cDNA was obtained in a DNA polymerase I system. The double-stranded cDNA was purified using AMPure XP Beads (Beckman Coulter, Brea, CA, USA). The double-stranded cDNA was ligated to the sequencing adaptor after terminal repair and A tail, and 250–300 bp cDNA was obtained using AMPure XP beads. Finally, the PCR system was amplified, and the PCR products were purified again using AMPure XP beads to obtain the libraries.

Library quality was examined using the Agilent Bioanalyzer 2100 system. The effective concentration of the library ($\leq 2$ nM) was quantified using qRT-PCR. After passing the inspection, the libraries were pooled and sequenced on the Illumina HiSeq X-10 (Illumina, San Diego, CA, USA) platform by Beijing Novo Biotech Company, Ltd. Finally, each sample contained an average of 6.63 G of valid data, and $4.42 \times 10^7$ clean reads.

## Quality control

To ensure the accuracy of data analysis, we filtered the original data and examined the sequencing error rate. Using in-house Perl scripts to process the raw reads of fastq format. Removing reads containing adapters, ploy-N sequences, and low-quality from the raw data to obtain clean reads. The Q20, Q30, and GC contents of the clean data were calculated. All subsequent analyses depend on clean data, high quality.

## RNA-seq data analysis

The analysis and calculation of all transcriptome data referred to a previous research report (*Trapnell et al., 2012*). In brief, the index of the reference genome was constructed using HISAT2 v2.2.1. The paired-end clean reads were obtained using HISAT2 v2.2.1 (https://cloud.biohpc.swmed.edu/index.php/s/fE9QCsX3NH4QwBi/download) aligned to the reference genome Zhongmu No. 1 (https://figshare.com/articles/dataset/genome_fasta_ sequence_and_annotation_files/12327602) to obtain mapped reads (*Mortazavi, Williams & McCue, 2008*). We also analysed the proportion of mapped reads in the exons, introns, and intergenic regions of the genome.

The clean reads aligned to Zhongmu No. 1 were quantified using FeatureCounts v1.5.0-p3. Gene expression was represented as FPKM (fragments per kilobase of transcript per million fragments mapped), and differences between WL 712 and Aohan FPKM values were compared using FeatureCounts v1.5.0-p3. Differential expression analysis of the two comparison combinations was performed using the DESeq2 R package (1.16.1) (https://www.bioconductor.org/packages/release/bioc/html/DESeq2.html). DESeq2 determines the differential expression in digital gene expression data using a model based on a negative binomial distribution. The corrected $P$-values and |log2foldchange| are thresholds for significant differential expression. $P$-values were adjusted using the Benjamini & Hochberg method.

Gene Ontology (GO) (http://www.geneontology.org/) enrichment and KEGG (Kyoto Encyclopedia of Genes and Genome (http://www.genome.jp/kegg/) statistical analysis of DEGs were performed using the clusterProfiler R package. A corrected $P$-value less than 0.05 was used as the threshold for significant enrichment of differentially expressed genes.

## qRT-PCR

The accuracy of the RNA-seq was verified by qRT-PCR. Total RNA was isolated from stems, and cDNA was synthesised by using the PrimeScript RT reagent Kit (Takara, Tokyo, Japan). Alfalfa $\beta$-Actin 2 was used as the internal reference gene. The primers in Table S2 were used for qRT-PCR. qRT-PCR was completed using the LightCycler 96/LightCycler480 system. The solution of the 20 µL system contained 0.4 µL forward primer, 0.4 µL reverse primer, 10 µL TB Green Fast qPCR Mix (2X) (Takara, Tokyo, Japan) and 2 ng cDNA. The PCR procedure included 45 cycles, with three technical repetites for each reaction. According to Kenneth report, the relative expression of each gene was calculated (*Livak & Schmittgen, 2001*).

## Statistical analysis

All statistical analysis was using *SPSS* software (version 17; IBM Inc, USA). The data were compared using Student's $t$-test, and $P < 0.05$ was considered statistically significant. The power of our samples was calculated using RNASeqPower (https://bioconductor.org/packages/release/bioc/html/RNASeqPower.html), and the RNASeqpower was 94.2%.

## RESULTS

### Phenotypic analysis of five alfalfa varieties

To compare the differences in the growth patterns of the five cultivars (Table S1A), plant height, internode length and stem diameter of alfalfa at different growth stages were continually measured in 2019 and 2020 (Fig. 2, Table S3). There were no significant differences in plant height, internode length or stem diameter among cultivars at the seedling transplant stage. After the budding stage, plant height, internode length and stem diameter of different alfalfa varieties reached a plateau and remained relatively stable (Figs. 2A–2C). In 2019 and 2020, WL 712 and Aohan represented tall and short phenotypes, respectively (Fig. 2D). Comparing the agronomic traits of alfalfa at the budding stage in

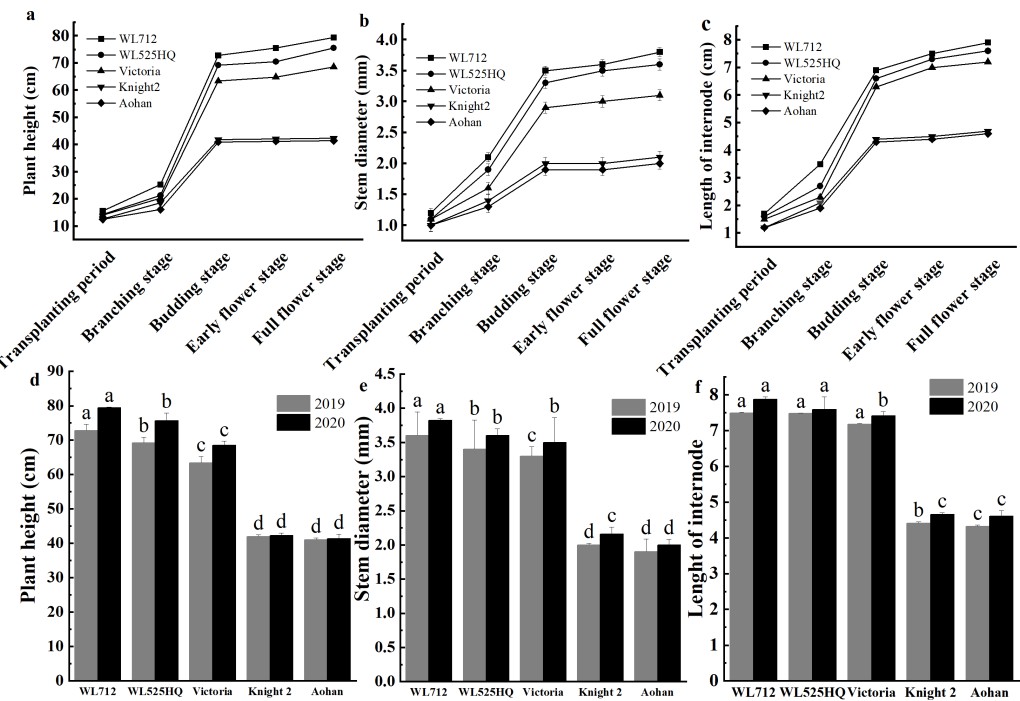

**Figure 2** **Phenotypic evaluation of five alfalfa cultivars.** The dynamics of plant height (A), stem diameter (B) and internode length (C) of five alfalfa cultivars during the indicated stages of development. Here, we recorded the transplanting stage as 0 day, branching stage (18 d), budding stage (42 d), early flower stage (45 d) and full flower stage (50 d). Average plant height (D), stem diameter (E), internode length (F) of five alfalfa cultivars. The values are the average of fifteen biological replicates and error bars represent the standard deviation. Different letters indicate significant difference at $P < 0.05$ among the five cultivars as determined by Student's t test.

2019 and 2020, the plant height of WL 712 was approximately 1.78 and 1.91 times those of Aohan, respectively, and the stem diameter of WL 712 was approximately 1.90 and 1.92 times those of Aohan (Figs. 2D–2E). The internode length and number of lateral branches in WL 712 were significantly larger than those in Aohan ($P < 0.01$), whereas the number of main branches in WL 712 was significantly lower ($P < 0.05$) (Fig. 2F, Figs. 3A–3B).

To identify the correlation between internode length and stem diameter and other traits, the fresh weight, leaf-stem ratio, and dry weight of the five cultivars were also determined. The results showed that the production performances of WL 712 and Aohan were significantly different ($P < 0.05$) (Figs. 3C–3F). Phenotypic correlation analysis based on 8 agronomic traits was done. We found that fresh and dry weight were positively and strongly correlated with the number of lateral branches, plant height, stem diameter, and internode length, and plant height was significantly positively correlated with internode length ($P < 0.01$). In addition, the number of main branches was negatively correlated with plant height, stem diameter, and internode length ($P < 0.01$) (Table 2).

From the screening of five alfalfa cultivars, WL 712 and Aohan were identified as the cultivars with the most significant difference in growth performance (Fig. 1). The growth trend of the two varieties in greenhouse is similar to that in field. The plant height, internode

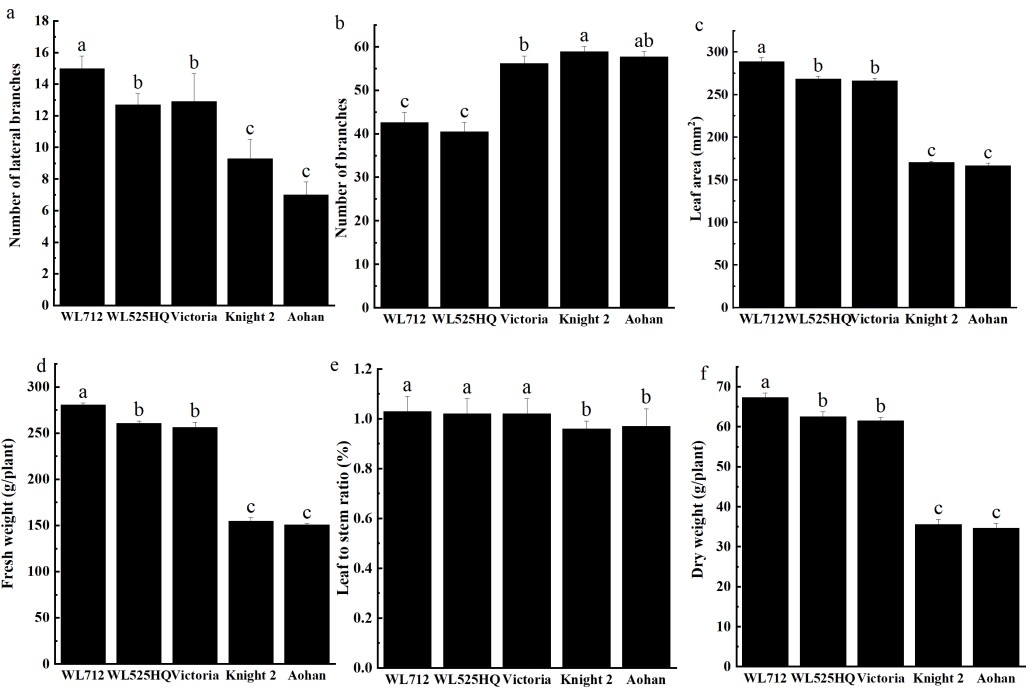

**Figure 3  Phenotypic evaluation and index determination of five alfalfa cultivars at budding stage (42 d).** Lateral branch number (A), total branch number (B), leaf area (C), fresh weight (D), leaf to stem ratio (E) and dry weight (F) of five alfalfa cultivars. The values are the average of fifteen biological replicates and error bars represent the standard deviation. Different letters indicate significant difference at $P < 0.05$ among the five cultivars as determined by Student's t test.

length, yield per plant, leaf area and stem diameter of WL 712 alfalfa were significantly higher than those of Aohan alfalfa (Table 2).

Based on the above results, WL 712 and Aohan were used as the vigorous-growing and slow-growing experimental cultivars. A piece of the stem, midway between stem tip and base, of plants at the budding stage (approximately 42 days after transplanting) was used for RNAseq.

## RNA-seq analysis

Using RNA-seq, we obtained $2.74 \times 10^8$ raw reads. The sequence error rate of a single base position was 0.03%, and the average GC content was 41.65%. After filtering from the raw data, $2.65 \times 10^8$ (96.94%) clean reads (39.76 G) were obtained. The phred values were greater than 97% and 93% at Q20 and Q30, respectively (Table S1C). The Pearson coefficient showed that the homology among the samples within the group was higher than 84.6% (Fig. S1).

We aligned the clean reads with the reference genome. The average proportions of exons, introns and intergenic regions in AJ samples were 72.72%, 3.61%, and 23.67%, respectively. Similarly, the WJ samples accounted for 74.14%, 2.96%, and 22.90%, respectively (Table S4). The reads aligned to the intron region may have been derived

| | PH | SD | IL | LBN | MBN | FW | LSR | DW |
|---|---|---|---|---|---|---|---|---|
| PH | 1 | | | | | | | |
| SD | 0.98** | 1 | | | | | | |
| IL | 0.99** | 0.98** | 1 | | | | | |
| LBN | 0.89** | 0.92** | 0.90** | 1 | | | | |
| MBN | - 0.84** | −0.76** | −0.79** | −0.68** | 1 | | | |
| FW | 0.98** | 0.98** | 0.98** | 0.91** | −0.75** | 1 | | |
| LSR | 0.55** | 0.54** | 0.55** | 0.51** | −0.46* | 0.60** | 1 | |
| DW | 0.99** | 0.99** | 0.99** | 0.82** | −0.76** | 1.00** | 0.59** | 1 |

**Table 2** Correlation coefficients between traits among the five alfalfa cultivars.

**Notes.**

*, ** Significant at the 0.05, and 0.01 probability levels, respectively.

PH, plant height; IL, internode length; SD, stem diameter; FW, fresh weight; LSR, leaf-to-stem ratio; DW, dry weight; LBN, lateral branch number; MBN, main branch number.

from the precursor mRNA. The reads aligned to the intergenic region may have been derived from ncRNAs.

Additionally, according to the comparison of RNA-seq data from WL 712 and Aohan, the RNASeqpower of our sample was 94.2%. The result may be beneficial to screen and explore the functional DEGs related to the vigorous-growing of alfalfa. These results demonstrated that the experiments were reproducible and that the data were accurate.

## Identification and functional annotation of DEGs in WL 712 and Aohan

Generally, the gene expression value of RNA-seq is evaluated as fragments per kilobase of transcript per million mapped reads (FPKM), which corrects the sequencing depth and gene length (Fig. S2). More than 90% of the clean reads were successfully mapped to the alfalfa genome. To clarify the function of the DEGs between WL 712 and Aohan, we performed GO and KEGG enrichment analyses. In total, 954 DEGs were significantly enriched and assigned to 35 GO terms. Compared to Aohan, WL 712 up-regulated 578 genes and down-regulated 376 genes. Among the molecular functions, "*protein heterodimerization activity*" (GO:0046982) (114 DEGs, 11.95%) had the highest proportion, followed by "*UDP-glycosyltransferase activity*" (GO:0008194) (99 DEGs, 1.04%) and "*translation factor activity, RNA binding*" (GO:0008135) (86 DEGs, 9.01%). Among the cell components, "*bounding membrane of organelle*" (Go:0098588) (57 DEGs, 5.97%) represented the largest cluster, followed by "*whole membrane*" (Go:0098805) (49 DEGs, 5.13%) and "*peptidase complex*" (Go:1905368) (44 DEGs, 4.61%). Among the biological processes, "*translational elongation*" (GO:0006414) (41 DEGs, 4.30%) represented the largest cluster (Table 3, Table S5, Fig. 4).

Based on biological system network, the function of DEG was identified by using KEGG classification. A total of 1324 genes were enriched in 110 KEGG pathways (Fig. 5). "*Carbon metabolism*" (ath01200) (103 DEGs, 7.8%) and "*Ribosome*" (ath03010) (96 DEGs, 7.3%) were the most abundant pathways, followed by "*Biosynthesis of amino acids*" (ath01230) (81 DEGs, 6.1%), "*RNA transport*" (ath03013) (54 DEGs, 4.1%), "*Plant-pathogen interaction*" (ath04626) (52 DEGs, 3.9%), "*Protein processing in endoplasmic reticulum*" (ath04141) (52
**Table 3** Top 10 gene ontology function classification.

| Category | Description | GO ID | Count | Percentage (%) |
|---|---|---|---|---|
| **Biological process** | translational elongation | GO:0006414 | 41 | 4.30 |
| | regulation of protein complex disassembly | GO:0043244 | 8 | 0.84 |
| | regulation of translation | GO:0006417 | 7 | 0.73 |
| | regulation of translational elongation | GO:0006448 | 7 | 0.73 |
| | regulation of translational termination | GO:0006449 | 7 | 0.73 |
| | translational frameshifting | GO:0006452 | 7 | 0.73 |
| | posttranscriptional regulation of gene expression | GO:0010608 | 7 | 0.73 |
| | positive regulation of cellular protein metabolic process | GO:0032270 | 7 | 0.73 |
| | regulation of cellular amide metabolic process | GO:0034248 | 7 | 0.73 |
| | positive regulation of cellular amide metabolic process | GO:0034250 | 7 | 0.73 |
| **Cell component** | bounding membrane of organelle | GO:0098588 | 57 | 5.97 |
| | peptidase complex | GO:1905368 | 44 | 4.61 |
| | whole membrane | GO:0098805 | 49 | 5.13 |
| | proton–transporting two–sector ATPase complex | GO:0033177 | 20 | 2.10 |
| | Golgi apparatus part | GO:0044431 | 36 | 3.77 |
| | Golgi apparatus | GO:0005794 | 36 | 3.77 |
| | proteasome core complex | GO:0005839 | 37 | 3.88 |
| | COPI–coated vesicle membrane | GO:0030126 | 9 | 0.94 |
| | COPI–coated vesicle | GO:0030137 | 9 | 0.94 |
| | COPI vesicle coat | GO:0030126 | 9 | 0.94 |
| **Molecular function** | translation elongation factor activity | GO:0003746 | 41 | 4.30 |
| | translation factor activity, RNA binding | GO:0008135 | 86 | 9.01 |
| | acid–amino acid ligase activity | GO:0016881 | 12 | 1.26 |
| | UDP–glycosyltransferase activity | GO:0008194 | 99 | 1.04 |
| | threonine–type endopeptidase activity | GO:0004298 | 37 | 3.88 |
| | threonine–type peptidase activity | GO:0070003 | 37 | 3.88 |
| | protein heterodimerization activity | GO:0046982 | 114 | 11.95 |
| | ligase activity, forming carbon–nitrogen bonds | GO:0016879 | 35 | 3.67 |
| | acetylglucosaminyltransferase activity | GO:0008375 | 35 | 3.67 |
| | oxidoreductase activity | GO:0016638 | 12 | 1.26 |

DEGs, 3.9%) and "*Plant hormone signal transduction*" (ath04075) (44 DEGs, 3.2%) (Table S6).

## Expression and regulation of DEGs in WL 712 and Aohan

KEGG analysis showed that DEGs related to stem elongation and diameter enlargement were widely involved in biological processes such as hormone signalling, photosynthesis and transcriptional regulation (Table S7).

Plant hormone signal transduction (Ath04075) involves many hormones that regulate growth and development, such as auxins, cytokinins, gibberellins, brassinosteroids, jasmonic acid, and ethylene. Twelve DEGs were enriched in the auxin-mediated signalling pathway, including *auxin-responsive protein SAUR* (*SAUR*), *auxin-induced protein X10A* (new gene), and *auxin transporter-like protein* (*LAX*). Among these, *IAA9, IAA6, SAUR50,*

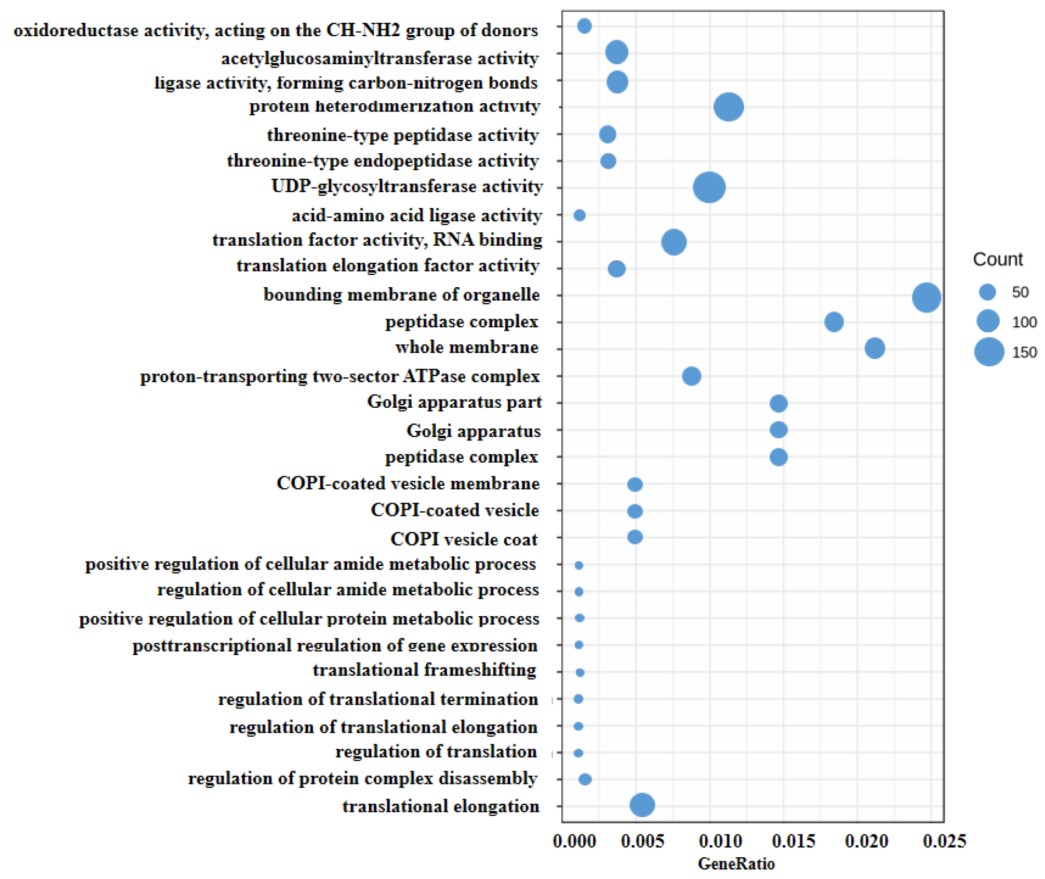

**Figure 4** **Scatter diagram of enriched GO functional categories.** The "GeneRatio" shows the ratio of the number of DEGs in the given category to the total number of differentially expressed genes. The size of the spot indicates the approximate number of DEGs in the category, all the spots indicate the significance level, $P < 0.05$.

*SAUR32*, and *SAUR36* were significantly up-regulated. In the cytokinin-mediated signalling pathway, four DEGs encoded enzymes, such as *adenylate isopentenyltransferase 5 (IPT5)*, *7- deoxyloganetin glucosyltransferase (UGT85A24)*, *cytokinin dehydrogenase 6 (CKX6)*, and *cytokinin hydroxylase (CYP735A2)*. *DELLA protein GAI (GAI)*, *f-box protein GID2 (GID2)*, and *transcription factor PIF4 (PIF4)* were enriched in the gibberellin-mediated signalling pathway. *Serine/threonine-protein kinase BSK8 (BSK8)*, *serine/threonine-protein kinase BSK1 (BSK1)*, and *cyclin-D3-3 (CYCD3-3)* were enriched in the brassinosteroid-mediated signalling pathway. Five DEGs were enriched in the jasmonic mediated signalling pathway, including *Coronatine-insensitive protein homolog 1a (COII A)*, *protein TIFY6B (TIFY6B)*, *protein TIFY11B (TIFY11B)*, *protein TIFY10B (TIFY10B)*, and *protein TIFY3B (TIFY3B)*. Four upregulated DEGs were enriched in the ethylene-mediated signalling pathway, including *ethylene receptor (ETR1)*, *mitogen-activated protein kinase kinase4 (MKK4)*, *mitogen- activated protein kinase homolog MMK1 (MMK1)*, and *protein ethylene insensitive 3 (EIN3)*.

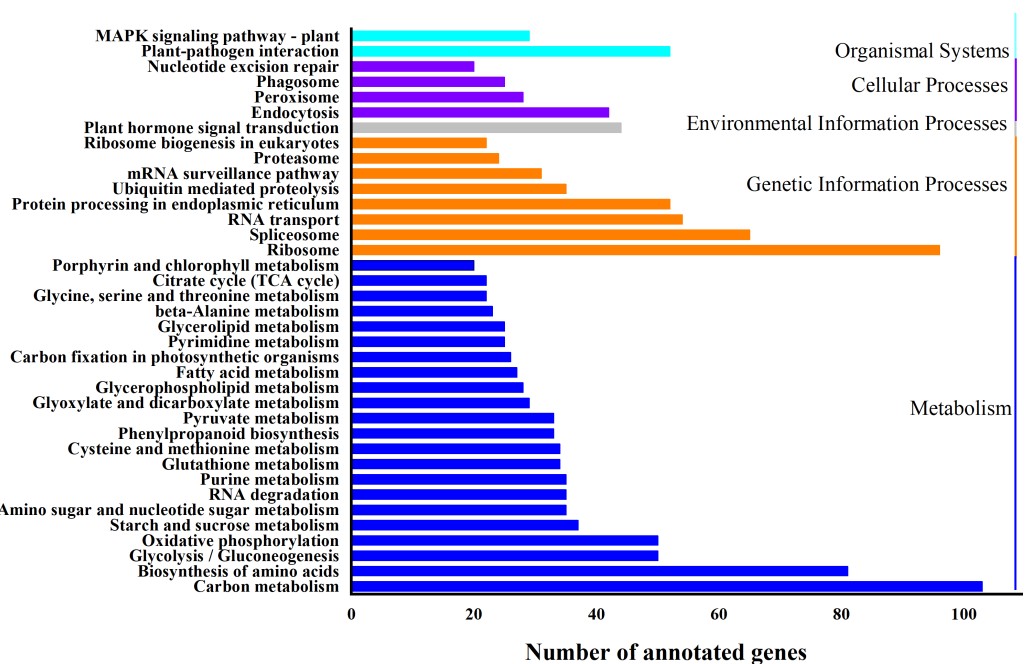

**Figure 5** **KEGG classification of differentially expressed genes (DEGs).** X-axis is the number of gene annotations; *Y*-axis is the type of KEGG pathway.

Fifteen DEGs were enriched in the photosynthetic pathway (ath00195). Among them, *PPL1*, *PETC*, *PSBR*, *PSBS*, *PSAG*, *PSAO*, *PSB27*, and *PSB28* were related to the photoreaction. *PLSN2* was related to the activity of the chloroplast NAD(P)H dehydrogenase (NDH) complex. *ATPF2* and *ATPC* are related to ATPase activity. Additionally, two oxygen-evolving enhancer proteins and ferredoxins have been identified. In the photosynthesis-antenna protein (ath00196) pathway, eleven DEGs were classified into *chlorophyll a-b binding proteins* and *chlorophyll a/b binding proteins*, which were expressed in chloroplasts. In the MAPK signalling (ath04016) pathway, twenty-two DEGs were mainly involved in biotic stress (pathogen infection), abiotic stress (cold/salt/drought/osmotic stress), and hormone synthesis during root growth and wounding responses.

Furthermore, the TCA cycle (ath00020), carbon fixation in photosynthetic organisms (ath00710), glycolysis/gluconeogenesis (ath00010), ribosome (ath03010), amino sugar and nucleotide sugar metabolism (ath00520), pyruvate metabolism (ath00620), and phenylpropanoid biosynthesis (ath00940) appeared closely related to alfalfa growth (Table S7).

In the TCA cycle pathway, 12 DEGs are annotated as playing a role in catalysis of the pyruvate dehydrogenase complex. In addition, *ATP-citrate synthase alpha chain protein 1* (*ACLA1*) and *2 malate dehydrogenases* (*MDH*) were identified. *Pyrophosphate-fructose 6-phosphate 1- phosphotransferase subunit beta* (*PFP*) and *glycoaldehyde-3-phosphate dehydrogenase* (*GAPC1*), both members of the glycolysis/gluconeogenesis pathway, were highly expressed in WL 712. Seven *glyceraldehyde-3-phosphate dehydrogenase* (*GAPDH*)

genes were enriched in the "carbon fixation in the photosynthetic organism" pathway and were highly expressed in the fast-growing cultivar. Ribosomal proteins predominated in the ribosomal pathway and included proteins that are parts of the 30S subunit (*RPS1*, *RPS13*, *RPSQ*, *RPS16*), 40S subunit (*RP24a*, *RP30a*, *RP15d*, *RP10a*, *RP20a*), 50S subunit (*RPL28*, *RPMJ*, *RPL31*, *RPLX*), and 60S subunit (*RPP3a*, *RPL21e*, *RPL37a*, *RPL37B*). *Dihydrolipoyllysine-residue acetyltransferase component 2 of the pyruvate dehydrogenase complex* (putative ortholog of *ODP22*) and *malate dehydrogenase* (*mMDH*) were highly expressed in WL 712 among the pyruvate metabolic pathway genes. *Beta-glucosidase 44* (*BGLU44*), *beta-amylase 1* (*BAM1*), *acid beta-fructofuranosidase* (*VCINV*), and *probable fructokinase-4* (*SCRK4*) were highly expressed in WL 712 among genes in the starch and sucrose metabolism pathways. The genes the phenylpropanoid biosynthesis pathway with high expression were *probable cinnamyl alcohol dehydrogenase* (*CAD2*), *beta-glucosidase 46* (*BGLU46*), *trans-cinnamate 4-monooxygenase* (*CYP73A3*), and *3 peroxidases* (*PER*).

## DEGs enriched in a variety of biological processes

All DEGs were analysed using GO and KEGG analyses. We found seven groups of DEGs plausibly related to stem elongation and diameter expansion, including formation of water-conducting tissue in vascular plants, cell division and shoot initiation, biosynthesis and degradation of lignin, cell enlargement and plant growth, formation of the primary or secondary cell wall, cell elongation, and stem growth and induced germination (Table S8). Fourteen DEGs were enriched in lignin biosynthesis and degradation. Peroxidases play an important role in this process. Additionally, *peroxidase 47* (*PER47*) is a novel gene (Fig. 6A). Eleven DEGs were enriched in the formation of the primary or secondary cell wall class. *Cellulose synthase A catalytic subunit* (*CESA*) is up-regulated in WL 712 (Fig. 6B). Eighteen DEGs were enriched in the cell enlargement and plant growth category. AUX IAA proteins, such as *auxin-responsive protein* (*IAA9*), *auxin-induced protein* (*IAA6*), and *auxin transporter-like protein* (*LAX5*), were particularly abundant. Additionally, *auxin-induced protein X10A* is a novel gene (Fig. 6C). Five DEGs were enriched in the cell division and shoot initiation category. Genes encoding the enzymes such as *7- deoxyloganetin glucosyltransferase* (*UGT85A24*), *cytokinin hydroxylase* (*CYP735A2*), and *cytokinin dehydrogenase 6* (*CKX6*) were particularly abundant (Fig. 7A). Two DEGs were enriched in the stem growth and induced germination category. Interestingly, one them, *DELLA protein* (*GAI*), is argued to negatively regulate the gibberellin signalling pathway, whereas the other, *F-Box protein* (*GID2*), is supposed to regulate that pathway positively (Fig. 7B). *Serine/threonine-protein kinase (BSK1)* and *BSK8* are thought to be related to cell elongation (Fig. 7C). *Eukaryotic translation initiation factor 5A-1* (*EIF5A*), *mitogen-activated protein kinase kinase kinase 3* (*ANP3*), and *alpha, alpha-trehalose-phosphate synthase* (*TPS6*) are apparently involved in the formation of water-conducting tissues (Fig. 7D). Additionally, we identified genes that are thought to regulate senescence, including *protein ethylene insensitive 3* (*EIN3*) (Fig. 6D). Importantly, compared with Aohan, *cellulose synthase A catalytic subunit 8* (*CESA8*), *beta-1,4-xylosyltransferase* (*IRX9*), *probable beta-1,4-xylosyltransferase* (*IRX14H*), *auxin-responsive protein* (*SAUR36*), *peroxidase 16* (*PER16*), and *peroxidase 51* (*PER51*) were upregulated more than 8-fold in WL 712, whereas *mitogen-activated protein kinase 3*

(*MPK3*), *pathogenesis-related protein* (*PR-1*), *peroxidase 55* (*POD55*), *beta-glucosidase 46* (*BGLU46*), and *peroxidase 15* (*POD15*) were down-regulated more than 15-fold in WL 712 (Table S9). All the genes that might be related to stem growth and development were clustered together, as shown in Figs. 6 and 7.

## Transcription factors potentially involved in alfalfa growth and development

Transcription factors are essential in plant growth and development as protein molecules that regulate gene expression. In this study, 20 transcription factors were implicated in the difference between fast and slow growing alfalfa cultivars (Fig. 8A, Table S10). Seven DEGs were upregulated, including *NAC domain-containing protein 73* (*NAC073*), *NAC domain-containing protein 10* (*NAC010*), *transcription factor MYB 46* (*MYB46*), and *NAP-related protein 2* (*NRP2*). Additionally, *WRKY transcription factor 22* (*WRKY22*), *transcription factor TGA 1* (*TGA1*) and *transcription factor MYB86* were novel genes. GO annotations state that *NAC073* and *NAC010* are involved in the synthesis of cellulose and hemicellulose and the development of secondary cell wall fibres. Thirteen DEGs were down-regulated, and the *WRKY* and *MYB* family members were conspicuous among them. GO classification state that *WRKY51* is involved in the positive regulation of salicylic acid-mediated signal transduction and negative regulation of jasmonic acid-mediated signal transduction in the defense response. *WRKY54* is apparently a negative regulator of plant growth and development. *MYB46* is apparently involved in secondary wall cellulose synthesis as a transcriptional activator. Finally, *MYB86* is apparently involved in lignin synthesis and accumulation. Additionally, *MYB2* is known to inhibit the expression of light-harvesting genes. All identified transcription factors were validated using qRT-PCR (Fig. 8B). The relative expression of *NAC081* was significantly upregulated in WL 712 ($P < 0.001$). The relative expression levels of most transcription factors were similar to the FPKM trend.

### The reliability of RNA-seq was verified using qRT-PCR

To determine the accuracy and rationality of the data, we arbitrarily selected 11 DEGs for qRT-PCR validation. The chosen DEGs were mainly related to the formation of the primary or secondary cell wall, cell enlargement and plant growth, and synthesis and degradation of lignin.

The changes in transcript abundance are shown in Fig. 9A. Consistent with RNAseq, qRT-PCR revealed that *IRX9*, *CESA8*, *CESA7*, *MKK4*, *PER16*, and *PER51* were significantly upregulated in WL 712 ($P < 0.05$). *MPK3*, *PR-1*, *BGLU46*, and *POD15* were significantly down-regulated in WL 712 ($P < 0.05$) (Fig. 9B). However, the relative expression of *CAD2* between the two varieties was not significantly different ($P > 0.05$) and was inconsistent with the RNA-seq transcript abundance. This may have been caused by RNA-seq errors in the acceptable range. Overall, the relative expression trend of the DEGs was similar to the RNA-seq.

## DISCUSSION

Alfalfa is an important component of feed, and the growth performance of its aboveground part affects the biomass yield. The *FmS6K* gene plays an important role in regulating the

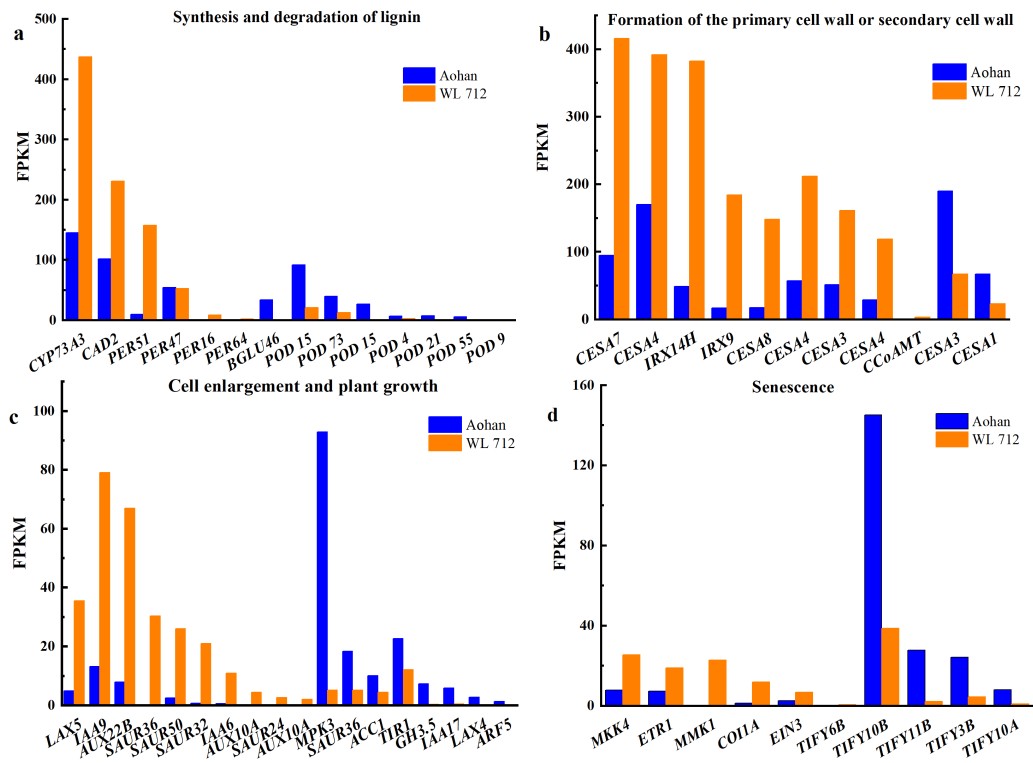

**Figure 6** Bar graphs showing the FPKM (fragments per kilobase of transcript per million mapped reads) of DEGs involved in various biological processes distinguished by GO enrichment analysis. (A) Synthesis and degradation of lignin; (B) formation of the primary cell wall or secondary cell wall; (C) cell enlargement and plant growth; (D) senescence.

development of plant stems (*Sun et al., 2018*). The yield of elephant grass has a strong positive correlation with internode length (*Yan et al., 2021*). However, the molecular regulatory mechanisms underlying the growth rate of stems and branches in alfalfa remain unclear. In this study, the growth difference between the tall and fast growing variety WL 712 and the short and slow-growing variety Aohan was studied. The transcriptome of those two varieties was analyzed by RNA-seq, with RNA obtained from the mid region of the stem. The difference between qRT-PCR and RNA-seq of individual DEGs may be caused by the error of RNA-seq within the acceptable range. Overall, the RNA-seq data could be used for subsequent analysis.

All DEGs were associated with at least one GO term; 954 significant DEGs were obtained, and seven DEG clusters were speculated to be involved in promoting fast growth (Figs. 6, 7).

Additionally, KEGG revealed that hormone signal transduction, photosynthesis and phenylpropanoid biosynthesis genes are up-regulated in the faster growing cultivar. RNA-seq also identified several novel DEGs associated with the fast growing cultivar, including *PER47* and *TIFY10A*.

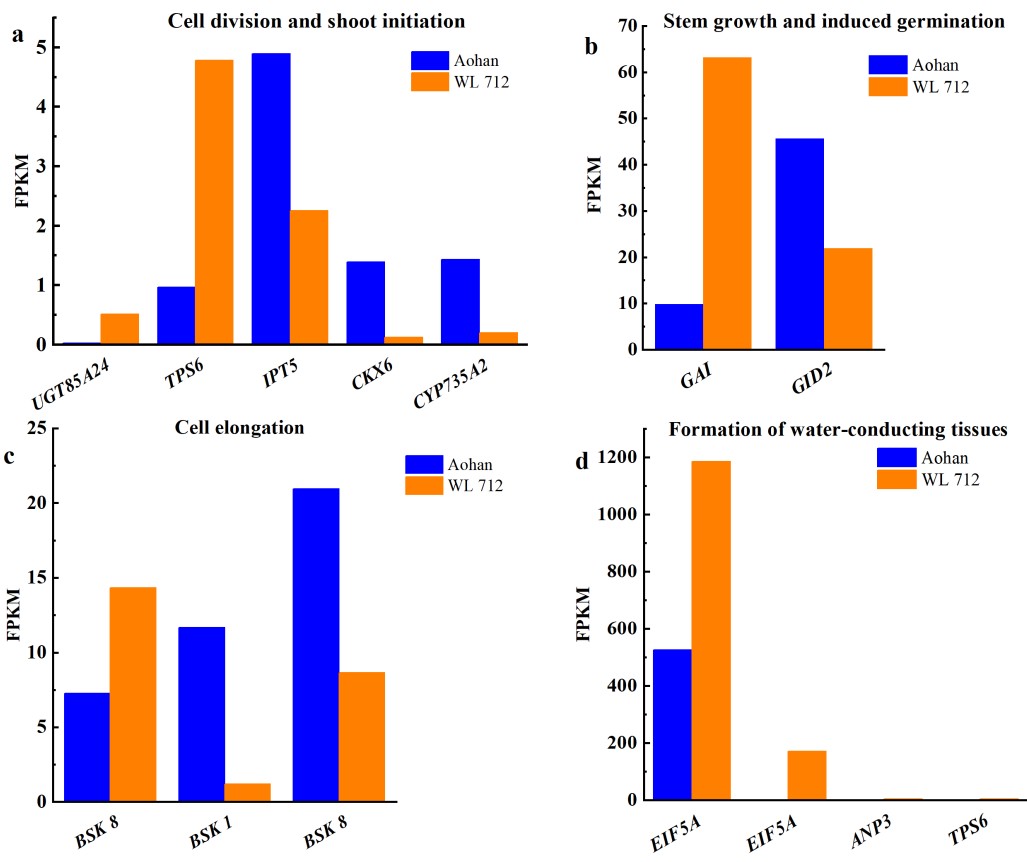

**Figure 7** **Bar graphs showing the FPKM of DEGs involved in additional biological processes distinguished by GO enrichment analysis.** (A) Cell division and shoot initiation; (B) stem growth and induced germination; (C) cell elongation; (D) formation of water-conducting tissues.

Plant organ growth is influenced by both developmental processes and environmental factors (*Sun et al., 2018*). In many cases, these changes are due to hormone-mediated action (*Verma, Ravindran & Kumar, 2016*). Here, auxin, cytokinin, gibberellin, ethylene, brassinosteroid, and jasmonic acid were all implicated because their downstream targets were found among DEGs, such as *SAUR50*, *CKX6*, *GID2*, and *GAI*. These DEGs might play a role in promoting fast growth in alfalfa. Previous studies have identified *SAURs* as a class of hormones that regulate plant growth and development and promote cell enlargement (*Ren & Gray, 2015*). Cytokinin synthesis is required to activate shoot division in apple trees with the top removed (*Tan et al., 2018*). Relevant studies have shown that gibberellin regulates plant organ elongation and development (*Nagel, 2020*). *GAI* is an inhibitor of highly conserved gibberellin signalling in plants. The *SCF* (*GID2*) complex mediates degradation of DELLA proteins (*RLG2*, *RGA*, and *GAI*), and activates and positively regulates the gibberellin signalling pathway (*Dill et al., 2004*). In addition, in the plant hormone signal transduction pathway, the production of hormones that play a mediating role depends on the metabolism of amino acids or fatty acids. Tryptophan in plants is not only involved in the synthesis of proteins but also the precursor of many metabolites (such

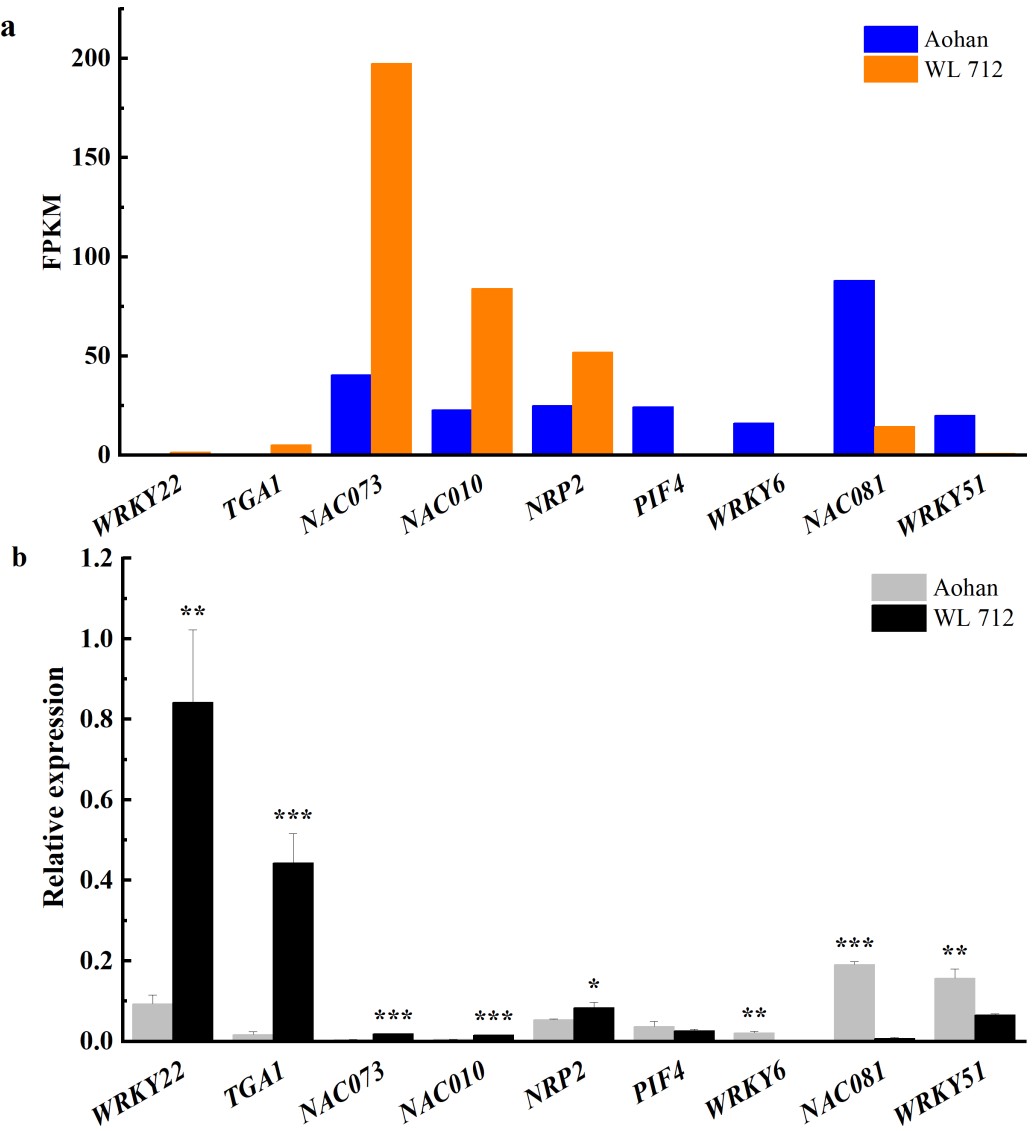

**Figure 8** **Transcription factors putatively involved in stem elongation and diameter enlargement in alfalfa as distinguished by GO analysis.** (A) Bar graphs showing the transcript abundance based on FPKM. (B) Bars plot the relative expression levels based on qPCR. Asterisks (*, **, ***) indicate expression level of the cultivars is significantly different at the 0.05, 0.01, and 0.001 probability levels, respectively. The expression levels of all genes are plotted relative to the expression level of the internal standard ($\beta$-*Actin*).

as auxin) (*Mano & Nemoto, 2012*). Jasmonic acid induces plants to prioritise defense over growth by interfering with the gibberellin signalling cascade, which is usually accompanied by significant growth inhibition (*Yang et al., 2019*). *TIFY*, which encodes jasmonic acid repressor, was significantly upregulated in Aohan (Table S7). This may explain why Aohan alfalfa is a dwarf plant.

Photosynthesis is an essential metabolic process. Twenty-nine DEGs were related to photosynthesis. For example, *PIF1* and *PIF3* were significantly down-regulated in WL

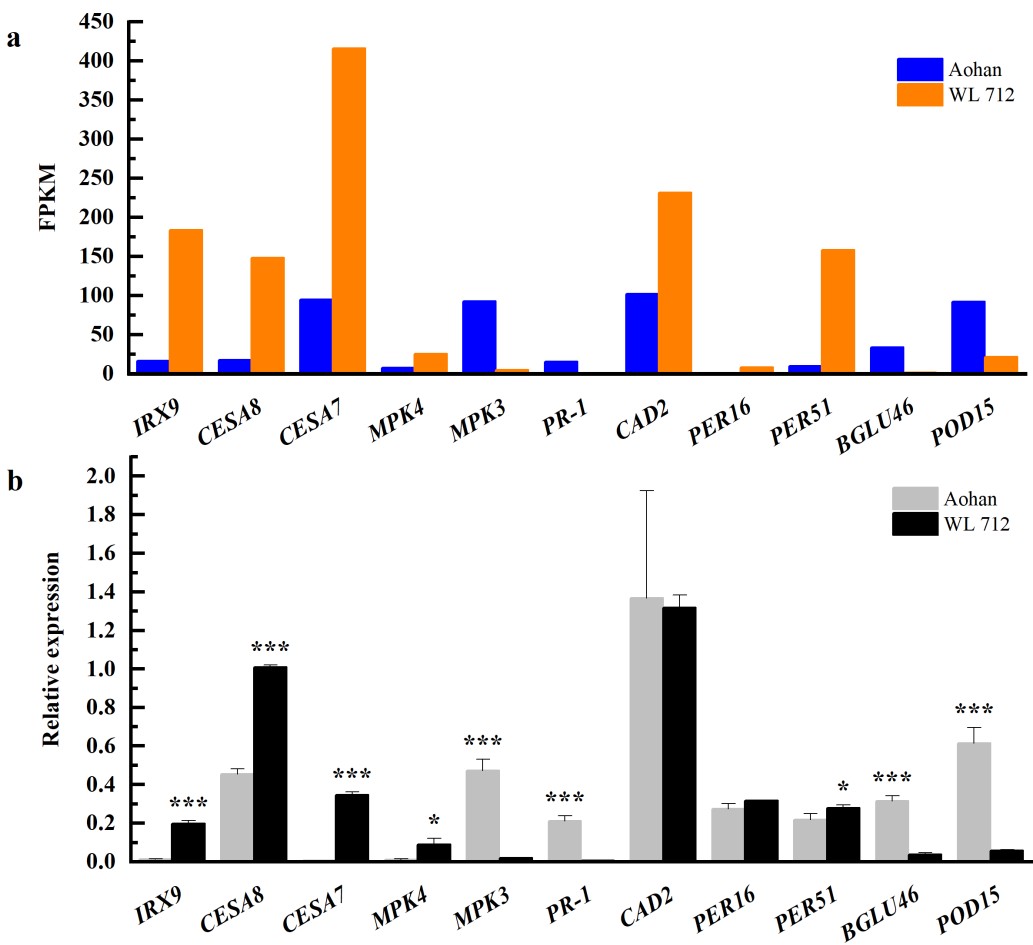

**Figure 9** **Comparison of RNA-seq and qRT-PCR for 11 genes.** (A) RNAseq bars show transcript abundance based on FPKM. (B) qRT-PCR bars show transcript abundance based on qRT-PCR. Asterisks (*, ***) indicate expression level of the cultivars is significantly differene at the 0.05, and 0.001 probability levels, respectively. The expression levels of all genes are plotted relative to the expression level of the internal standard ($\beta$-Actin).

712 (Table S7). These genes may play a regulatory role in the process of plant height and internode elongation. Plant height and leaf area of transgenic soybean are decreased by overexpressing *PIF4* (*Arya, Singh & Bhalla, 2021*). The deletion of *PIF1* and *PIF3* results in an increase in plant height, longer internodes, and late flowering (*Hoang et al., 2021*). The light-harvesting complex II (LHC II) functions as a light receptor and is related to the absorption of light (*Gu et al., 2017*; *Sen et al., 2021*). The up-regulation of LHC II DEGs may enhance the photosynthesis of WL 712 and promote the growth of plants. Additionally, circadian rhythm is also involved in the regulation of plant growth and development (*Venkat & Muneer, 2022*). Our research found that DEGs enriched in circadian rhythm pathway were mainly related to photoperiod flowering response (Table S7).

RNA-seq analysis found 1531 DEGs related to rape stem growth (*Yuan et al., 2019*). Combined analysis of proteome and RNA-seq found that DEGs and DEPs of *Mikania*
*micrantha* stems were significantly enriched in photosynthesis, carbon sequestration, and plant hormone signal transduction pathways (*Cui et al., 2021*). We identified seven DEG clusters that were plausibly involved in stem elongation and enlargement. Fourteen DEGs were annotated as being involved in lignin synthesis and degradation and peroxidases (Fig. 6A).

These genes may regulate lignin synthesis and degradation in stems. The oxidation activity of peroxidases is important for lignification (*Hoffmann et al., 2020*). Eleven DEGs were apparently involved the formation of the primary or secondary cell wall (Fig. 6B), with good representation from cellulose synthase. Previous studies reported that *CESA 4* and *CESA8* are specifically enriched and expressed in the stem tissue during the fiber development stage (*Guo et al., 2021*). Eighteen DEGs were enriched in the category of cell enlargement and plant growth frequently involving auxin (Fig. 6C). Five DEGs were apparently involved in cell division and shoot initiation (Fig. 7A). Two DEGs were enriched in the categories of stem growth and induced germination (Fig. 7B), mainly components of the gibberellin signalling pathway. Two DEGs were potentially involved in cell elongation (Fig. 7C). These DEGs might play a role in stem internode elongation, diameter enlargement and lateral branch formation. Previous studies reported that *AtTPS6* completely compensates for the defects in reduced trichome and stem branching due to *CSP-1* deficiency in *Arabidopsis thaliana* (*Chary et al., 2008*). Deletion of *IAA17* in tomatoes showed that the increase in fruit size is related to the higher ploidy level of peel cells (*Su et al., 2015*). Finally, the *TIFY* homologs possibly involved in senescence were also identified here (Fig. 6D). In addition, we identified several members of *SPL* family, such as *SPL1*, *SPL6* and *SPL7*, which may be involved in the lateral branch development of alfalfa.

Previous studies reported that *SPL13* regulates shoot branching in alfalfa (*Gao et al., 2018*). Overall, these DEGs may be involved in alfalfa growth and development. Transcription factors are essential in the regulation of development, morphogenesis and responses to environmental stress. Previous research found that most members of *NAC*, *WRKY* and *MYB* families are involved in the synthesis of lignin, cellulose, and hemicellulose (*Wang et al., 2016*). The *NAC*-mediated transcription network synergistically regulates synthesis of the plant secondary wall (*McCarthy, Zhong & Ye, 2011*). *WRKY6* and *WRKY33* positively regulated abscisic acid signal transduction during early development of *A. thaliana* (*Huang et al., 2016*).

*WRKY54* is a negative regulator of salicylic acid synthesis (*Li, Zhong & Palva, 2017*) and can significantly increase stem diameter, leaf area, and total dry weight of plants (*Amin et al., 2013*). Overexpression of *AtMYB44* in tomatoes results in slow growth (*Shim et al., 2012*). *MYB3R1* is a transcriptional repressor that regulates organ growth, and restricts plant growth and development by binding to target genes and promoters of specific genes (*Wang et al., 2018*). Under reduced light intensity, *MYB2* and *MYR1* act as inhibitors of flowering and organ elongation, respectively (*Zhao et al., 2011*). Here, excluding *WRKY22*, all *WRKY* members were significantly upregulated in dwarf alfalfa. Therefore, *WRKY22* may positively regulate the growth and development of WL 712. *NACs* are involved in the development of plant secondary cell walls. Among these, *NAC081* functions as a positive

regulator. *MYB46* and *MYB86* might positively regulate the synthesis of cellulose and lignin, and *MYB44*, *MYB3R1* and *MYB2* might act as transcriptional repressors (Table S10).

## CONCLUSION

Plant height is an important factor in determining forage biomass. The molecular characteristics of the DEGs between fast and slow growing alfalfa cultivars were identified using RNA-seq. The trend of our qRT-PCR was largely consistent with those of RNA-seq, which indicated that the RNA-seq data could be used for subsequent analysis. All DEGs were analysed using GO terms, and 954 significant DEGs were identified. KEGG analysis indicated that hormone signal transduction, phenylpropanoid biosynthesis, and photosynthesis are well represented in the fast growing cultivar. GO analysis highlighted the following seven clusters of DEGs: formation of water-conducting tissue, cell division and shoot initiation, synthesis and degradation of lignin, stem growth, formation of the primary or secondary cell wall, cell enlargement and plant growth, and induced germination and cell elongation. Additionally, the transcription factors implicated in stem elongation and diameter expansion are mainly *WRKY*, *NAC*, and *MYB* family members. In summary, our research results not only enrich the transcriptome database of alfalfa, but also provide valuable information for explaining the molecular mechanism of fast growth, and can provide reference for the production of alfalfa around the world.

### Funding
This work was supported by the China Agriculture Research System of MOF and MARA. The funders had no role in study design, data collection and analysis, decision to publish, or preparation of the manuscript.

### Grant Disclosures
The following grant information was disclosed by the authors:
China Agriculture Research System of MOF and MARA.

### Competing Interests
The authors declare there are no competing interests.

### Author Contributions
- Jiangjiao Qi conceived and designed the experiments, analyzed the data, prepared figures and/or tables, authored or reviewed drafts of the article, and approved the final draft.
- Xue Yu performed the experiments, prepared figures and/or tables, and approved the final draft.
- Xuzhe Wang performed the experiments, prepared figures and/or tables, and approved the final draft.
- Fanfan Zhang performed the experiments, prepared figures and/or tables, and approved the final draft.

- Chunhui Ma conceived and designed the experiments, authored or reviewed drafts of the article, and approved the final draft.

## Data Availability

The raw data are available in the Supplemental Files. The sequences are available at NCBI: PRJNA807394.

## Supplemental Information

Supplemental information for this article can be found online at http://dx.doi.org/10.7717/peerj.14096#supplemental-information.

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
