# Peer review of "Differentially expressed genes related to plant height and yield in two alfalfa cultivars based on RNA-seq"

_PeerJ, doi:10.7717/peerj.14096_

## Round 0.1 · original submission · Major Revisions

Thank you for submitting your paper to Peer J. Three reviewers have read it and generally find the results worthwhile. Reviewers 1 and 2 have a variety of minor points that I am sure you can address. However, reviewer one points to some more substantial issues. I have read your paper myself and find problems, which I will elaborate here.

First there is a technical matter. You measure growth phenotypes on plants grown in the field. However, the plants used for RNAseq were made from cuttings and grown in a greenhouse. Thus, the datasets come from plants grown under distinct conditions and for this reason are difficult to compare. This unusual growth protocol needs to be explained to the reader but more importantly, how do the growth phenotypes documented for field grown plants resemble those grown from cuttings in the greenhouse. Without data on this point, I don’t think the transcriptomic data are interpretable.

Next is a conceptual matter. You wish to assign function to the identified differentially expressed genes. To be sure, the two genotypes (WL712 and Aohan) differ in growth intensity and some of those genes are presumably related to these phenotypes. But in addition, there is genetic difference between the genotypes and the distinct expression will also reflect the two lineages. In its present form, I think it is difficult to determine whether any given differentially expressed gene relates to the phenotypes or to the divergence between the lines.

There are various ways in which helpful data could be obtained. For example, the Knight 2 cultivar resembles Aohan in its growth behavior and Victoria (and WL525HQ) resembles WL712; by using two genotypes for the fast growing type and two others for the slow, a stronger selection criterion could be imposed (relevant genes would have to be differentially expressed between pairs of genotypes). The usefulness of this method depends on the genetic distance between these cultivars, but that should be determinable. Reviewer 1 suggests getting transcriptomic data for all of the cultivars studied phenotypically and of course that would be even better.

Reviewer 1 also suggests obtaining transcriptomic data at several time points. The reason for this is because of the substantial circadian regulation of many transcripts. Doing so would certainly enrich the paper because you would be able to identify those genes differentially regulated because of circadian regulation. I am not certain this is essential, but if left out then the potential interference from circadian regulation should be discussed.

Finally, there is a second conceptual matter. What is the justification for focusing exclusively on stem height and diameter? You never answer this question. In the introduction, there is a large paragraph from lines 50 to 75 that has information about alfalfa growth and performance. In this section, buried in the middle (lines 60 to 64), you refer to stem height and diameter but you never make a case for why these two phenotypes are so important. In fact, within those lines, you tell us that the number of branches and also the duration over which growth takes place is important. Indeed, it is not hard to come up with other parameters: leaf area, root size, nodulation, among others. Looking at stem length and width seems arbitrary.

Based on the data presented, WL712 grows more vigorously than Aohan; that seems clear. But you also show that along with stem length and width, many other phenotypes are correlated. It is not at all surprising that plants with bigger leaves should have bigger stems and more biomass. But maybe the larger leaves drive stem growth, instead of the other way around. I suggest that you take a step back and interpret you data more generally in terms of the difference between vigorous growth (WL712) and slower growth (Aohan). Alternatively, if indeed stem growth is to be the focus then you need to justify this choice.

Because taking care of the above will probably involve new experiments, you are likely to need more than 35 days, even double or triple or that. That is fine, but if so please the journal know that you do plant to resubmit.

Other issues

The hormonal data are misinterpreted. While there might be statistical differences in the level of the hormones, the small absolute size of the differences means that they are biologically irrelevant. As a rule, plants respond to the logarithm of the hormone level; but in any case, the differences seen were not even two-fold. Given the differences in genotype and plant size, these measured differences are simply not meaningful. In fact, the correct conclusion is that whatever accounts for the differences between growth for the two genotypes, it is *not* hormone concentration. However, I caution that plant hormones are difficult to measure. I saw no efforts to validate the assays used. The data on hormone levels might not even be reliable.

In today’s journals, a figure may have many panels but all of the panels for a single figure number fit on one page. The current Figure 1 spreads over three pages. This should be renumbered to be Fig 1, 2, and 3, or redrawn so that all of the panels fit on one page.

The discussion is long and rambling. Remember that transcription is regulated loosely and therefore that the differential expression of many, perhaps most, genes matters little if at all. These kinds of RNAseq datasets can provide hints to function, but no more. In contrast to the many claims in the paper, transcript levels in themselves demonstrate nothing about function. Your discussion is entirely speculative. The value here is a dataset for alfalfa that interested parties can search for their own interests. There is no need for prolonged discussion. Pick the most important things and comment succinctly. Perhaps two or three pages only.

The bibliography is a mess. One of the reviewers commented that bibliographies are supposed to be presented in alphabetical order based on the surname of the first author. In biology, we only write out surnames (family name) in full, other names are given as initials. Unfortunately, the conventions for bibliographies were developed in the west without regard to Chinese names. For a non-westerner it can be difficult to tell which is the family name. One way to tell is to look at published bibliographies (in pdf versions of papers). Find other papers that cite the papers you are citing and see how the names are handled. There are many other kinds of mistakes in the bibliography. Please have someone go over your bibliography who is knowledgeable about formatting practices or pay an agency to do so.

Reviewer 1 ·

Basic reporting

Qi et al. have conducted a very interesting comparative study on alfalfa cultivars with the aim of discovering differentially expressed genes (DEGs) that might be related to plant height and yield. The authors have conducted a transcriptomic analysis along with phenotypic evaluations of the cultivars at different developmental stages. While the phenotypic portion of the study is interesting and comprehensive, the transcriptomic portion needs major improvement. I respectfully describe some of the important changes that need to be made to this study before publication.

Experimental design

1- The authors have provided phenotypic data for five cultivars, but transcriptomic data for only two cultivars (AJ and WL). For a comprehensive transcriptomic analysis, it is important to include, in the transcriptomic experiments, all the other cultivars included in the phenotypic analyses.

2- Transcriptomic analyses are usually done at different timepoints to avoid detecting false DEGs, which could be due to daily changes in gene expression. I recommend conducting transcriptomic analyses for each cultivar at different timepoints, e.g. every 8 hours within 24 hours.

Validity of the findings

While, the DEGs identified in this study are interesting, it is unclear whether they are truly related to the plant height and yield phenotypes. A more comprehensive study design, see previous section, would help make the discovered DEGs more reliable.

Additional comments

1- References need to be organized either alphabetically or with numbers. Currently, the references do not seem to be in any specific order.

2- Several important references related to this study should be cited and discussed. Here are some examples:

a. https://bmcgenomics.biomedcentral.com/articles/10.1186/s12864-016-3014-6

b. https://www.nature.com/articles/s41598-018-27088-8

c. https://idp.springer.com/authorize/casa?redirect_uri=https://link.springer.com/article/10.1007/s12298-021-01026-x&casa_token=s75y7baZ7mkAAAAA:RaGawp1aShEnhwCSPKRhwgEqLIddg7n4Au4RUq0OOzCV7J4vsdgHZKZThWvK-sVB2EQJ93IV2shMFwB7


3- All scripts for this study, including the ones used for RNAseq and the statistical comparisons, need to be provided in a public repository, for example github.

Minor comments:
1- In line 72, it is unclear what country is referred to by "at home".

2- In line 77, “nonparameteric transcriptomic analysis” should be defined inside the sentence.


3- In line 86, Arabidopsis thaliana should be italicized.

4- In line 524, the link to SRA is incorrect and leads to submission page rather than the public page for the project. The correct link is https://www.ncbi.nlm.nih.gov/sra/?term=PRJNA807394


5- In figure 2 , to avoid confusion, the significant difference between "WJ" and "AJ" for each treatment should be shown with asterisks rather than letters. Letters are often sued for pairwise comparison, i.e. comparing different cultivars for the same treatment.
6- In figure 4, bar graphs are shown, but the caption calls them “histograms”. I suggest changing "histogram" in the figure caption to "bar graphs". This is also true for figure 5a
7- Also in figure 4, the results for 8 biological processes are presented but the caption says "DEGs involved in seven biological processes".
8- In the caption of figure 5, FPKM should be written in uppercase.

Reviewer 2 ·

Basic reporting

1. Line no. 21 Specify which variety refers to the results mentioned in this part. The Result part seems to discuss the RNA-seq data in general. Include the results generated from qRT-PCR conducted to validate the generated RNA-seq data.
2. Be consistent in writing the term “RNA-Seq” or “RNA-seq” in the whole manuscript, including the keyword section. “High throughput” vs “High-throughput”, “Fpkm” vs “FPKM”, and “P-value” vs “P value” are other terms that are not written consistently. This should not exist in scientific writing. Be critical and meticulous.
3. Line no. 43. Check typos in the whole manuscript.
4. Line no. 78. Describe Zhongmu 1. Is it an alfafa cultivar?
5. Make sure the scientific names are italicized, i.e., Arabidopsis thaliana.
6. Line 118-120 Needs to be rephrased and described clearly.
7. Line 480. Include reference.
8. Table S3 is not mentioned in the manuscript.
9. Some labels are missing in Figure 3b. For example, does it correspond to the same KEGG classification?
10. The gene names in Figure 4 need to be italicized. Make sure the gene names in the whole manuscript are italicized too.

Experimental design

Experimental design is sufficient.

Validity of the findings

The conclusion needs to summarize the findings comprehensively. The qRT-PCR results need to be discussed and conclude whether the results correlate with the obtained RNA-seq.

Additional comments

Too many inconsistencies in writing several terms as mentioned earlier. Be critical and meticulous in scientific writing.

Reviewer 3 ·

Basic reporting

Overall, it is a nice written manuscript
1: The authors claim to have filled the gap in knowledge, but the data they use is old. Please clarify what new work in this direction has been done in terms of increasing alfa-alfa biomass.

Experimental design

Experimental design is well laid out except my query Comment 2

Validity of the findings

2: Line 107-108: Alfa-alfa is one of the most important forage crops grown in the cold, arid trans-Himalayan region, where the soils are naturally deficient in many essential and microelements. Justify your addition of all minerals to the soil and your collection of data. There ought to have been a check of alfalfa growth in natural soil. This can strengthen your claim.
3: Line 123,129: Justify why the particular stem size was selected?
4: Alfa Alfa is a herbaceous perineal crop. Do you have any data regarding seasonal change in its growth?
5: While the overall conclusion is not very appealing to the audience, can you translate this to
other varieties of the same species growing elsewhere in the world?

Additional comments

Conclusion needs revision its not catching my attention

---

## Round 0.2 · Minor Revisions

Thank you for resubmitting your paper to PeerJ. As both reviewers note, the paper is improved. From my own perspective, I still consider the study to be weak. I mentioned in my previous letter that because you study only two cultivars you have no way to know whether the reason for the differential gene expression is based on the phenotype (i.e., fast growth) or the genotype (i.e, lineage). Also, your developmental time course shows that plant traits at budding stage are the same as at the later stages. Therefore you don’t know whether the faster growth of WL712 is because the tissues expand faster or because the period over which growth occurred was longer. If the latter is true, then the difference between the cultivars would be timing not actual growth rate and your RNA would be comparing plants that had just stopped growth to plants that had stopped growth some time ago. All of these problems make it difficult to say very much about the role of the genes you found.

Having said that, I recognize the value of the transcriptomic data. Despite the limitations, others might be able to mine it for useful information. Therefore I am prepared to accept the paper. However, there are small issues to attend to. Reviewer one makes a few small but important points including that you upload your scripts.

I also read your paper carefully and edited directly on the pdf. Most of these edits are small matters of English. I put a few comments on the text as well. I won’t repeat those here. Please use your pdf viewer to show you the list of edits and comments and make sure you change those places, either as indicated or in some way you think is appropriate. If you don’t understand a particular edit, by all means email me directly.

Many of the edits were needed to tone down your claims. Remember that GO and KEGG categories are simply guesses. The guesses might be reasonable but they are guesses all the same. But even if we accept that the GO category is correct, there are many reasons why a gene might be a DEG in your study that have nothing to do with regulating fast growth. Accordingly, I added words like ‘apparently’ to indicate that the connection is suggested but not shown.

Here are a couple of further comments

Paragraph starting line 93. Here and elsewhere, you use ‘fast-growing’ (or ‘vigorous-growing’) in a way that confuses me. You write “the fast-growing of alfalfa”. Actually, ‘fast growing’ is an adjective. A person might write about a “fast-growing” variety. By contrast, it is strange to write ‘the fast-growing of a plant’. Do you mean ‘growth rate’? Certainly the ‘growth rate’ of a plant would be an important factor affecting its yield. But maybe there is some special alfalfa thing where ‘fast growing’ has a defined meaning? A particular stage? On the text, I have edited with ‘growth rate’ because this seems reasonable. But if there is a more specific meaning then more editing will be required.

Because this is central to your paper I will pursue the language further. Let’s consider cars. Suppose we have a Trabant and a Ferrari. We might be interested in why one car goes so much faster than the other. But we would not write ‘the fast moving of the Ferrari’. We could write about the ‘rapid speed’ of that car. But the general question would about speed itself, what determines the speed? That is the general question (fast or slow). In principle it could be just as interesting to learn why Aohan grows slowly as WL712 grows fast.

Notice above, I used the word ‘determines’, not ‘regulates’. The word ‘regulates’ has some issues. Again thinking about a car, what regulates the speed of the car? My foot on the gas pedal does. If I want the car to go faster, I push down my foot on the pedal. But how about the gas in the tank? Does the gas ‘regulate’ the speed of the car? No, the gas is burned, it is needed but it is not regulating the speed. What if we found out that one car goes faster than another because that car has higher quality gas? That would be a case where the gas influences, affects, or even determines the car’s speed. But the word regulate implies a control process, like a person’s foot on the pedal.

Line 225. Please give the age of the plants at budding stage (presumably number of days in the greenhouse. Also, was that the same for both cultivars? It would be useful for the legend of figure 2 a-c to add the days so we know roughly about the time intervals.

Supplemental Figure 1 is nice. Consider adding to the main text.

Figure 1
For a – c, difficult to distinguish Aohan and WL712 symbols. Also, the line colors are confusing. Try using the same color for the line and the symbol and making WL712 black?

For a, c, d, and f, delete the decimal point and the trailing zero for the y-axis labels (thus 1, 2, 3, not 1.0, 2.0, 3.0). Then, make the axis name larger so it is easier to read.

Figure 2. Again, remove trailing zeros (a, c, d, f) and make the y-axis names much bigger.

Please make figure 3a its own figure (Figure 3) and figure 3b its own figure (Fig. 4). And renumber the later figures accordingly.

Figure 4 (a – d). Please add a y-axis and tick marks to all panels. In panel b, the y axis has too many numbers (just have a number every 100).

Also for figure 4, please explain what FPKM stands for in the legend.

Finally for Figure 4, either place all panels a – h together on one page, or make two figures, each with its own number.

For figure 5 and, please use the same color scheme for the bars as used in figure 4. Also I am confused by the legend. You talk about ‘databases’ but what does that mean? Not from your work here? Please rephrase those legends.

When you re-submit please submit a version with track changes so that I can see how you changed the ms.

Reviewer 1 ·

Basic reporting

significantly improved

Experimental design

More or less the same as before, but removing the hormone assays

Validity of the findings

Same as the previous version

Additional comments

The manuscript has improved significantly compared to the previous version. I am happy to recommend this manuscript for publication, but there are still a few things that need to added or fixed:

1. The scripts (all codes used for the different analyses) have not been provided. I think for the sake of reproducibility, the authors should provide those scripts on Github. In their rebuttal letter as well as in the manuscript, the authors provide the link to the raw RNAseq data, but not the scripts.

2. The links to Hisat2 and the reference genome (Zhongmu No. 1) are incorrect and need to be fixed.

3. The caption of Figure 6a, still refers to the bar graphs as "histograms" which need to be fixed. On the same caption, "Fpkm" should be changed to "FPKM".

Reviewer 3 ·

Basic reporting

Satisfied with all my queries

Experimental design

Satisfied with all my queries

Validity of the findings

Satisfied with emy queries

Additional comments

Satisfied with all my queries

---

## Round 0.3 · Minor Revisions

Thank you for returning your manuscript promptly and with good attention to the comments. The paper is nearly ready. I saw no need to send it back to the reviewers, you have clearly satisfied their requests. However, I read it carefully and there are still a some English problems and a few small matters of science. Again, I made edits directly on the manuscript which I have uploaded for your attention. Please make sure that you look at the list of changes because a few of them are small deletions. Easy to miss. If you have questions about any of the changes by all means email me directly.

As for the small matters of science, although I have marked them with comments on the manuscript, I will go through them here too.

Line 115. “thirty-six stems with well-growing single alfalfa” makes no sense. Do you mean: "36 well growing, single stems were collected from each cultivar." If so, say that. If not, please clarify what you mean.

Line 137. Please state what part of the stem the region for RNAseq came from. Was it the basal-most part of the stem? The middle? The top? Since the total stem height was longer than the 1.5 cm sample used for RNA, you need to report carefully where the sampled part came from.

References. Several of the references appear to be incomplete. I have flagged them. Please check carefully. Also for your information, whenever a species is given in an article title by its scientific name (the Latin binomial), the genus is capitalized and the species is not; both genus and species go in italics.

Figures. In general the figures are much improved. I like the blue and orange bars. For Fig. 2, it would be helpful in the legend to give the plant age at each of those stages.

Figure 2f appears to be missing data.

Figure 6 and 7 need improved annotation (as specified in the note on the figure). Also in Fig. 6b, one of the genes is given by an Atg number. But that is an arabidopsis gene. I don’t see how that can refer to an alfalfa gene. Please rename.

Figure 8 and 9 in the legend should define relative. That is, what does a value of 1 mean.

---

## Round 0.4 · Minor Revisions

Thank you for your careful revision. I found only a small number of small mistakes. I have marked the manuscript accordingly. I am taking care with these small things because the journal does not have a copy editor and it is best to have as few mistakes as possible. When you resubmit, you do not need to comment on any of these small changes, unless I made a mistake myself, perhaps misunderstanding what you mean. As usual, it will be helpful for you to include a version of the manuscript with track changes shown. Almost there. Tobias Baskin

---

## Round 0.5 · accepted · Accept

Well done. I did find a few more small errors that I have corrected on the pdf (uploaded here). During the prduction process, you will have a chance to upload a version with these changes made. One change is perhaps larger than a simple mistake. For your title, the words at the start "Study of" have no content. Every research article is a "study of" what follows. So your title more powerfully starts "Differentially ..."

Thank you for submitting your paper to PeerJ. I hope you will be happy with the production and will recommend us to your friends.